# Using Deep Reinforcement Learning to Generate Rationales for Molecules

## Abstract

Deep learning algorithms are increasingly used in modeling chemical processes. However, black box predictions without rationales have limited used in practical applications, such as drug design. To this end, we learn to identify molecular substructures – rationales – that are associated with the target chemical property (e.g., toxicity). The rationales are learned in an unsupervised fashion, requiring no additional information beyond the end-to-end task. We formulate this problem as a reinforcement learning problem over the molecular graph, parametrized by two convolution networks corresponding to the rationale selection and prediction based on it, where the latter induces the reward function. We evaluate the approach on two benchmark toxicity datasets. We demonstrate that our model sustains high performance under the additional constraint that predictions strictly follow the rationales. Additionally, we validate the extracted rationales through comparison against those described in chemical literature and through synthetic experiments.

## 1 Introduction

Recently, deep learning has been successfully applied to the development of predictive models relating chemical structures to physical or biological properties, outperforming existing methods (Duvenaud et al., 2015; Kearnes et al., 2016). However, these gains in accuracy have come at the cost of interpretability. Often, complex neural models operate as black boxes, offering little transparency concerning their inner workings.

Interpretability plays a critical role in many areas including cheminformatics. Consider, for example, the problem of toxicity prediction. Over 90% of small molecule drug candidates entering Phase I trials fail due to lack of efficacy or due to adverse side effects. In order to propose a modified compound with improved properties, medicinal chemists must know which regions of the molecule are responsible for toxicity, not only the overall level of toxicity (Babine & Bender, 1997). We call the key molecular substructures relating to the outcome *rationales*. In traditional cheminformatics approaches such as pharmacophore mapping, obtaining such a rationale behind the prediction is an intrinsic part of the model (Martin et al., 1993; Durant et al., 2002; Katsila et al., 2016)

In this paper, we propose a novel approach to incorporate rationale identification as an integral part of the overall property prediction problem. We assume access to the same training data as in the original prediction task, without requiring annotated rationales. At the first glance, the problem seems solvable using existing tools. For instance, attention-based models offer the means to highlight the importance of individual atoms for the target prediction. However, it is challenging to control how soft selections are exploited by later processing steps towards the prediction. In this sense, the soft weighting can be misleading. In contrast, hard selection confers the guarantee that the excluded atoms are not relied upon for prediction. The hard selection of substructures in a molecule is, however, a hard combinatorial problem. Prior approaches circumvent this challenge by considering a limited set of predefined substructures (typically of 1-6 atoms), like the ones encoded in some molecular fingerprints (Durant et al., 2002). Ideally, we would like the model to derive these structures adaptively based on their utility for the target prediction task.

We formulate the problem of selecting important regions of the molecule as a reinforcement learning problem. The model is parametrized by a convolutional network over a molecular graph in which the atoms and bonds are the nodes and edges of the graph, respectively. Different from traditional reinforcement learning methods that have a reward function provided by the environment, our model

seeks to learn such a reward function alongside the reinforcement learning algorithm. More generally, our model works as a search mechanism for combinatorial sets, which readily expands to applications beyond chemistry or graphs.

Our iterative construction of rationales provides several advantages over standard architectures. First, sequential selection enables us to incorporate contextual features associated with past selections, as well as global properties of the whole molecule. Second, we can explicitly enforce desirable rationale properties (e.g., number of substructures) by including appropriate regularization terms in the reward function.

We test our model on two toxicity datasets: the Tox21 challenge dataset, which is a series of 12 toxicity tests, and the human ether-a-go-go-related gene (hERG) channel blocking. The reinforcement learning model identifies the structural components of the molecule that are relevant to these toxicity prediction tasks while simultaneously highlighting opportunities for molecular modification at these sites. We show that by only selecting about 40-50% of the atoms in the molecules, we can create models that nearly match the performance of models that use the entire molecule. By comparing selected regions with rationales described in chemical literature, we further validate the rationales extracted by the model.

## 2 RELATED WORK

**Deep Learning for Computing Chemical Properties**   One of the major shifts in chemical property prediction is towards the use of deep learning. The existing models fall into one of two classes. The first class of models is based on an expert-constructed molecular representation such as fingerprints that encapsulate substructures thought to be important and a range of molecular properties (Yap, 2011; Mayr et al., 2016). These models are not well suited for extracting rationales because desired structures may not be part of the fingerprint. Moreover, it may be challenging to attribute properties recorded in fingerprints to specific substructures in the molecule. One would have to restrict the feature space of the fingerprint, which can harm the performance of the model.

The second class of models move beyond traditional molecular fingerprints, instead learning task-tailored representations. Specifically, they employ convolutional networks to learn a continuous representation of the molecule (Duvenaud et al., 2015; Kearnes et al., 2016). Jin et al. (2017)'s work takes this a step further, and uses the Weisfeiler-Lehman kernel inspired neural model as a way to generate better local representations. Following this direction, our work is also based on learned molecular representation. Our focus, however, is on augmenting these models with rationales. As articulated in the introduction, the task is challenging due to the number of candidate substructures and the need to attribute properties aggregated in convolutions to individual atoms.

**Reinforcement Learning on Graphs**   Our work utilizes a similar framework as the reinforcement learning model over graphs described by Dai et al. (2017). However, their work focuses on solving computational problems where the reward is provided by the environment (i.e., a deterministically computable property of the graph, such as max-cut or the traveling salesman problem). In contrast, in our formulation, the rewards are also learned by the system.

Another related work in this area (Lee et al., 2017) utilizes a policy gradient method to search for substructures starting from random points in the graph. Since their model is designed for large graphs that do not fit into memory, it does not consider the global context of the molecule as a whole. This design is limiting in the context of molecular prediction, where such information is valuable for prediction, and where it is also feasible to compute since the graphs are small. Moving away from the convolutional network approach, their model imposes an artificial ordering of the nodes through a sequence network as the reinforcement learning agent traverses the graph. Both of the above models focus on the prediction task, whereas the emphasis of our work is on interpretability.

**Learning Rationales**   The topic of interpretability has recently gained significant attention (Letham et al., 2015; Kim et al., 2015). The proposed approaches cover a wide spectrum in terms of the desired rationales and the underlying methods. Work in this area include visualization of activations in the network (Karpathy et al., 2015; Hermans & Schrauwen, 2013; Li et al., 2015), and examination of the most influential data examples to provide interpretability (Koh & Liang, 2017). Attention-based models have also been widely used to extract interpretability (Ba et al.,

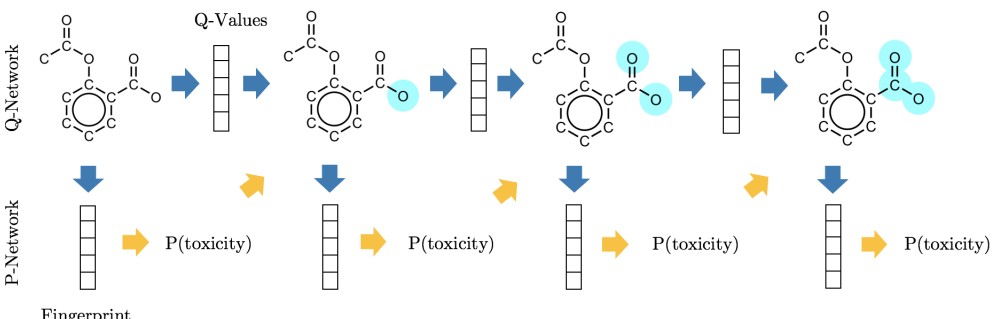

Figure 1: Our model makes sequential selections of atoms (light blue) in the molecule and is specified by two networks, the Q-Network and the P-Network. The former constitutes the reinforcement learning agent that assigns a Q-value to each atom, and the latter takes the atom selections of the Q-Network and trains a classifier to predict based solely on those atoms. This prediction is used as a reward that is fed back to the Q-Network.

2014; Cheng et al., 2016; Martins & Astudillo, 2016; Chen et al., 2015; Xu & Saenko, 2016; Yang et al., 2016). Our work is mostly closely related to approaches focused on extractive rationales (Lei et al., 2016; Ribeiro et al., 2016). Lei et al. (2016) present a model to extract parts of text as rationale, but their model does not readily generalize to graphs, and the sequential nature of our model can place a meaningful ordinal ranking over the atom selections.

## 3 MODEL

Our approach uses reinforcement learning as a method to iteratively select important atoms from the graph. We use a Q-learning approach similar to that of Dai et al. (2017). The state of the system at time $t$ corresponds to the atoms selected thus far. The agent takes an action $a_t$ at each time step $t$, where the action is the selection of an atom that has not already been selected. After the agent takes an action, the state $s_t$ is updated to include the newly selected atom. Unlike traditional reinforcement learning algorithms in which the agent receives a reward from the environment, we use a separate model (P-Network) to generate the reward $r_t$ to the agent. The P-network learns to predict molecular properties such as toxicity based on the selected atoms, and rewards the agent according to the accuracy of its predictions. The agent itself learns a Q-Network to represent the action-value function $Q(s, a)$ needed to maximize the reward throughout the process. In this case, maximizing the reward is equivalent to selecting atoms that help P-Network make accurate predictions based on the partial selections of atoms. The overall approach is illustrated in Figure 1.

In the following sections, we will describe the two networks, the Q-Network that guides the agent's actions, and the P-Network that maps agent's selections to predicted molecular properties. The two networks are trained iteratively, in a feedback loop, so that the Q-network can provide good selections of atoms that, when fed to the P-Network, result in good overall predictions of molecular properties. Both networks are parametrized by the graph convolutional network that we describe separately below.

### 3.1 CONVOLUTIONAL NETWORK

In a convolutional network, the model updates the feature representation of individual atoms iteratively based on local contexts. The nature of the operations is parameterized and can be learned to support the end-to-end task. We prefer convolutional networks due to their expressiveness and adaptability as compared to traditional molecular fingerprint methods.

Define a molecular graph as $G = (A, B)$, in which $A$ is the set of nodes denoting the atoms, and $B$ is the set of edges denoting the bonds between atom pairs. In each successive layer of the network, an atom's representation is updated to incorporate the currently available information from its neighbors, while the bond features remain unchanged. These updates propagate information

across the molecule, and allow the network to generalize beyond standard local substructures. Let $h_i^l$ be a vector-valued atom representation for atom $i$ at layer $l$, and let $A_i$ be the input feature vector for atom $i$, and $B_{i,j}$ be the input feature vector for the bond between atoms $i$ and $j$. In this notation, we use $h_i^0 = A_i$. (The input atom features include: one-hot encoding of the atomic number, degree, valence, hybridization, aromaticity, and whether or not the atom is in a ring. Bond features include: one-hot encoding of the bond order, aromaticity, and whether or not the bond is in a ring). This initialization differs slightly for the Q-Network so as to incorporate the current state of selections into the convolutional architecture. The update step for atom feature vectors involves a gated unit that receives separate contributions from the atom itself and its neighbors. Specifically,

$$ h_i^l = \sum_{j \in N(i)} \sigma \left( W_{g,nei}^l [h_j^{l-1}; B_{i,j}] \right) \cdot \left( W_{nei}^l [h_j^{l-1}; B_{i,j}] \right) + \sigma \left( W_{g,self}^l [h_i^{l-1}] \right) \cdot \left( W_{self}^l [h_i^{l-1}] \right) \quad (1) $$

where $N(i)$ is the set of neighbors of atom $i$, $W^l$'s are the specific weight matrices that vary across layers (bias weights omitted for brevity), and $\sigma$ is the sigmoid activation function, applied coordinate-wise. After $N$ iterations, we arrive at final atom representations $h_t^N$.

## 3.2 Q-Network

The Q-Network is parametrized by the convolutional network described in section 3.1, in which the size of the atom representation at the final layer is set to 1 so that $h_i^N$ is scalar and interpreted as the Q-value for selecting atom $i$ next. The initial atom features in the Q-Network include the binary indicator of whether the atom has already been selected. In other words, if we define the state $s$ as a binary vector encoding the atoms that have already been selected, then we use an augmented $h_i^0 = [A_i; s_i]$. The convolutional network is rerun with these initializations, using the same parameters, after each selection. Thus, despite the fact that the selections are greedy with respect to the Q-values, the model will choose atoms in a manner that is aware of the global context of the molecule as well as state $s$ representing already selected atoms.

## 3.3 P-Network

The P-Network is a prediction network that takes the partial selection of atoms from the Q-Network, and makes a prediction about the label of the molecule using only those atoms selected by the Q-Network. Like the Q-Network, the P-Network is separately parametrized by the convolutional network. To incorporate the selected atoms, we zero out all the initial atom features that are not in the current selection before running the convolutional model. It is important to note that we do not zero out the updated hidden states of these atoms. This allows interaction between disjoint selections on the molecular graph and preserves information related to graph distances. The reasoning behind this is that there are often several substructures of interest on the molecule and their interactions might prove important. We want to allow the network to learn these interactions by facilitating the propagation of information throughout the whole molecule.

The P-Network is geared towards predicting molecular properties rather than atomic properties and therefore requires an additional aggregation of the atom vectors $h_i^N$. In our model, these atom vectors are first renormalized via an attention mechanism:

$$ \hat{h}_i^N = \frac{e^{W_z h_i^N}}{\sum_j e^{W_z h_j^N}} h_i^N \quad (2) $$

and turned into a neural (adaptable) fingerprint by summing the resulting renormalized feature vectors $f = \sum_i \hat{h}_i^N$. This fingerprint is then passed through a sigmoid function to generate the class prediction, $\hat{y} = \sigma(W_f * f)$. The prediction loss is measured through standard cross-entropy loss $L_P = -y \log(\hat{y}) - (1 - y) \log(1 - \hat{y})$ where $y$ is the actual label and $\hat{y}$ is the predicted probability of $y = 1$.

## 3.4 Training

The reward for the Q-network is induced by the P-network and defined as:

$$ r_{t,t-1} = L_P(s_{t-1,\cdot}|\theta) - L_P(s_{t,\cdot}|\theta) \quad (3) $$

where $\theta$ refers to the parameters of the P-network and $s_{t,\cdot}$ is the binary state vector updated from $s_{t-1,\cdot}$ in response to action $a_t$. Because we are interested in selecting the important substructures of a molecule, which consist of at least several atoms, we found it useful to train the Q-network after $n$ selections rather than a single selection. The Q-Network is therefore trained using an $n$-step Q-learning algorithm, in which the network receives a signal for each sequence of $n$ actions that it makes. The loss function for the Q-network is then:

$$L_Q = (\gamma Q_{t+n}^{target} + r_{t+n,t} - Q_t)^2 \tag{4}$$

Where $\gamma$ is a decay constant, $Q_t$ specifies the Q-value of the current state and action, and $Q_{t+n}^{target}$ is the max Q-value of the state $n$ steps from $t$ induced by a separate but jointly trained target Q-Network.

During the training process, we use an $\epsilon$-greedy search policy, where the agent will choose a random atom with probability $\epsilon$ instead of the one with the highest Q-value. Keeping the idea of molecular substructures in mind, we find that it is helpful to search for a random neighbor of a selected atom, than a completely random atom. We also employ action-replay using a target Q network that utilizes soft parameter updates in order to increase training stability (Lillicrap et al., 2015).

In our model, we place limits on the numbers of atoms to be selected. This number is proportional to the number of atoms in the molecule up to a certain ceiling. We note that fixing the number of atoms chosen by the agent is a limitation of our model, but seems to work well in practice. Specifically, we find that taking 40-50% of the atoms up to a limit of 12-15 atoms for larger molecules works well, although this number varies with the problem. We also impose a lower limit of 5 atoms for smaller molecules, because it becomes impossible to distinguish distinct molecules when too few atoms are chosen.

Additionally, we impose regularization constraints on the model, in order to enforce certain properties on the selections. Since we are interested in the model selecting specific substructures from the molecule, we impose a penalty to the model for selecting too many disjoint groups. That is, we define a variable $C_t^g = \# \{\text{disjoint groups of atoms at step } t\}$. We then modify the reward equation 3 as follows:

$$r_{t,t-1} = L_P(s_{t-1,\cdot}|\theta) - L_P(s_{t,\cdot}|\theta) - \alpha(C_t^g - C_{t-1}^g) \tag{5}$$

## 4 Experimental Set-Up

**Datasets**   We evaluate our model on two toxicity datasets. The first dataset that we explore is the Tox21 challenge dataset which contains a series of 12 toxicity tests categorized by nuclear response signals and stress response pathways[1]. We parse the data using RDKit (Landrum, 2017), removing duplicate entries and cleaning the molecules by removing salts. Because this dataset has data coming from multiple sources, there are conflicting labels for some molecules, which we remove from the dataset entirely. Table 1 contains the information about each of the 12 datasets, highlighting the small number of positive examples in many of the datasets.

The second toxicity dataset that we evaluate our model on is the inhibition of the hERG channel[2]. Because this protein channel is well-studied, we explore this dataset to see if we can create a predictive model that can generate rationales that match the information in chemical literature. This dataset, taken from Li et al. (2017), consists of a training set with 3792 molecules, and a test set with 1090 molecules, with 25% positive labels. Since the data was already cleaned, we do no further preprocessing of this dataset.

**Evaluation Measures: Predictions**   For each dataset, we compare our model against the top reported systems Mayr et al. (2016); Li et al. (2017). These approaches utilize extensive feature engineering through molecular fingerprints and other computed molecular descriptors. In addition, they use additional training sources for data augmentation. Specifically, Mayr et al. (2016) utilize a data augmentation method called kernel-based structural and pharmacological analoging (KSPA), which uses public databases containing similar toxicity tests.

---

[1]https://tripod.nih.gov/tox21/challenge/
[2]https://ochem.eu/article/103592

We measure the predictive performance of the convolutional model (Mol-CNN, which utilizes the full molecule) to demonstrate that is comparable to the state-of-the-art results. Next, we evaluate the performance of our reinforcement learning method (RL-CNN) that makes predictions on a fraction of atoms in the molecule. We compare these different models using the AUC metric, since the datasets contain an unbalanced set of labels.

**Evaluation Measures: Rationales**   Because the RL-CNN model makes predictions using only atoms selected as rationales, its quantitative performance indirectly measures the quality of these rationales. However, we are also interested in directly evaluating their quality relative to rationales described in the chemical literature by domain experts.

In the ideal case, we would identify rationales that are characteristic to a single class of examples – either positive or negative. Unfortunately, many known toxic substructures are prevalent in both positively and negatively labeled compounds. In fact, Mayr et al. (2016) show that adding features representing structural similarity to 2,500 common toxicophores (toxic substructures) to their model does not improve performance on the Tox21 challenge dataset. This shows that the expert-derived "important" regions are not actually sufficient nor necessary for the prediction task.

Rationales extracted for the hERG dataset are directly compared with rationales described in the literature. Multiple studies have shown that the presence of a protonable nitrogen atom and 3 hydrophobic cores has a high affinity for binding to this particular protein (Cavalli et al., 2012; Sanguinetti & Tristani-Firouzi, 2006). Usually, this nitrogen is secondary or tertiary so that it is more basic. When protonated, the positive charge exhibits cation-pi interactions with certain residues of the hERG channel, which is the crux of the binding mechanism. We show that our model can identify these basic nitrogen atoms within the dataset.

For a baseline comparison, we also evaluate rationales obtained by selecting atoms with the strongest influence on the logistic regression model prediction using Morgan fingerprints of radius 3 and length 2048. Morgan fingerprints are boolean vectors constructed by enumerating atom-centered neighborhoods up to a certain radius, assigning integer values to each neighborhood by hashing a categorical feature vector of that neighborhood, and reducing those integers to vector indeces (Rogers & Hahn, 2010). The importance of an atom can be approximated by the absolute difference between a prediction made with the full molecular fingerprint and a prediction made when substructures containing that atom are excluded from the fingerprint (Riniker & Landrum, 2013). We restrict the baseline rationales to select the same number of atoms as in the RL-CNN model.

**Evaluation Measures: Rationales (Synthetic Experiment)**   Since we do not find well-defined substructures in literature for the tests offered in the Tox21 dataset, we also construct a synthetic experiment. For this experiment, we select specific substructures and set the labels of all molecules containing those substructures to be positive; all other molecules' labels are left unchanged. We specifically focus on 3 toxic substructures: the aromatic diazo group, polyhalogenation, and the aromatic nitro group, two of which are from Kazius et al. (2005)'s work on toxicophores common to multiple toxicity assays. We demonstrate that our model can capture important substructures if the data provides a clear enough signal.

## 5   RESULTS AND ANALYSIS

**Quantitative Evaluation of Convolutional Representation**   We first demonstrate that our convolutional model performs competitively compared to neural models using molecular fingerprints. Columns four and five in Table 1 compare our results for Mol-CNN with the highest performing model DeepTox (Mayr et al., 2016), both run in the multitask setting. The DeepTox model performs better than our convolutional model by 2.3% on average across the twelve tasks. This result is not surprising, as their method uses substantial amounts of additional training data which is not available to our model.

The results on the hERG dataset are summarized in Table 2. Our model outperforms the top performing model (Li et al., 2017), which uses molecular fingerprints, on the external test set.

**Quantitative Evaluation of Reinforcement Learning Algorithm**   For the Tox21 dataset, we use the multi-task instance to compare the results of our base convolutional model (Mol-CNN). How-

Table 1: Results of different models on the 12 Tox21 datasets using AUC as the metric. We run our base convolutional model (Mol-CNN) on the multi-task setting to compare with the top-performing system, DeepTox. To compare the results of our reinforcement learning algorithm (RL-CNN), we run our model in a single-task setting. Here, we report the results of the Mol-CNN, RL-CNN and a logistic regression (LR) baseline.

| Dataset | Size | % Pos Label | Multi-Task | | Single-Task | | |
| | | | DeepTox | Mol-CNN | Mol-CNN | RL-CNN | LR |
| --- | --- | --- | --- | --- | --- | --- | --- |
| NR-Ahr | 7302 | 11.1% | 92.3 | 90.1 | 89.5 | 89.3 | 87.1 |
| NR-AR | 8005 | 3.35% | 77.8 | 78.2 | 75.8 | 75.6 | 53.0 |
| NR-AR LBD | 7501 | 3.04% | 82.5 | 82.7 | 71.8 | 65.1 | 60.3 |
| NR-Aromatase | 6467 | 4.93% | 80.4 | 80.8 | 76.7 | 74.9 | 73.1 |
| NR-ER | 6757 | 10.5% | 79.1 | 77.2 | 73.1 | 73.6 | 70.5 |
| NR-ER LBD | 7714 | 4.14% | 81.1 | 74.5 | 68.2 | 73.1 | 81.2 |
| NR-PPAR-G | 7246 | 2.94% | 85.6 | 74.8 | 70.9 | 71.7 | 65.5 |
| SR-ARE | 6497 | 15.5% | 82.9 | 77.2 | 75.3 | 74.6 | 68.8 |
| SR-ATAD5 | 7859 | 3.86% | 77.5 | 79.2 | 75.6 | 74.7 | 69.6 |
| SR-HSE | 7232 | 4.95% | 86.3 | 84.3 | 79.1 | 83.5 | 75.5 |
| SR-MMP | 6481 | 14.8% | 93.0 | 93.7 | 93.2 | 90.7 | 86.9 |
| SR-p53 | 7571 | 6.16% | 86.0 | 83.5 | 80.4 | 75.1 | 67.4 |
| Average | - | - | 83.7 | 81.4 | 77.5 | 76.8 | 71.6 |

Table 2: Results of different models on the hERG dataset using AUC as the metric. The first 4 models are baselines from Li et al. (2017) and use molecular fingerprints as input to random forest (RF), support vector machine (SVM), k nearest neighbors (KNN) and associative neural networks (ASNN). For each of their models, we take the average performance for the same model run with different input features.

| Method | 5-Fold CV AUC | Test Set AUC |
| --- | --- | --- |
| RF | 69.2 | 68.2 |
| SVM | 71.4 | 74.7 |
| KNN | 83.2 | 75.8 |
| ASNN | 78.8 | 83.9 |
| Mol-CNN | 79.0 | 84.3 |
| RL-CNN | 78.7 | 80.9 |

ever, we turn to the single-task instance to evaluate the performance of our rationale model. This is due to the fact that different toxicity tests warrant different important substructures. Therefore, we run individual models for each of the toxicity tests using the base convolution model as well as the reinforcement learning model. We observe a small decrease in performance, resulting in a 0.7% decrease in AUC on average, using around 50% of the atoms in the dataset as seen in Table 1. On the hERG dataset, we selected 45% of the atoms, and also observe that the reinforcement learning algorithm performs similarly to the convolution network as seen in Table 2, with a 3.4% decrease in AUC. We see a smaller decrease in performance for the Tox21 datasets on average, likely because many of the datasets have comparatively few number of positive examples, so predicting on fewer atoms allows the model to generalize better.

**Evaluation of Rationales using Human Rationales**   In the absence of ground truth rationales, we turn to a specific structural motif–a tertiary nitrogen atom–that is known to exhibit cation-pi interactions with residues of the hERG protein when copresent with certain hydrophobic side chains in the correct 3-D conformation (Du et al., 2009; Cavalli et al., 2012). In the dataset we used, these tertiary nitrogen substructure occurs more often in positive examples compared to negative examples (78.4 % vs 44.9 %). This suggests that while this substructure is important in positive examples, it is not sufficient to indicate that a molecule is positive. We observe that our model captures this important substructure frequently, and more often in positive examples than negative

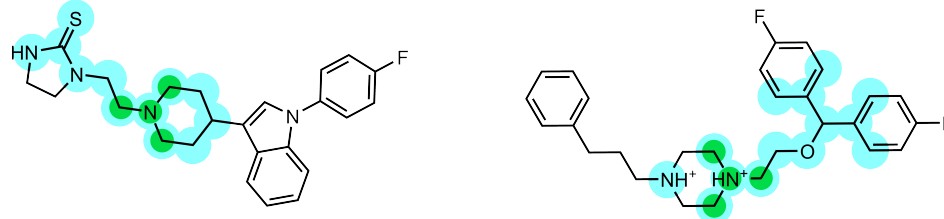

Figure 2: Two examples of rationales selected by the reinforcement learning model. The selected atoms are highlighted in large light blue circles. In both cases, we see that the model selects the tertiary nitrogen motif, highlighted in small green circles, which is implicated in many inhibitors of the hERG channel.

Table 3: Synthetic experiment results for the Tox21 challenge dataset. To test if our model is able to pick up specific signals when they are present, we altered training and test labels for molecules with known substructure (highlighted in green) to be always positive. The results show how well the model isolates these substructures as rationales for test molecules. The baseline LR model was evaluated analogously.

| Name | Structure | Total # Labels Altered | % Test Matches (RL-CNN) | % Test Matches (LR Fingerprint) |
|---|---|---|---|---|
| Aromatic Diazo | | 41 | 100% (5/5) | 60% (3/5) |
| Polyhalogenation | [Cl, Br, I, F] x 4 | 108 | 72.7% (8/11) | 45.5% (5/11) |
| Aromatic Nitro | | 250 | 76.5% (13/17) | 82.4 % (14/17) |

examples (63.6 % vs 46.1 %). Here, we require the model to have selected the nitrogen and at least two of its three carbon neighbors. Figure 2 shows two example selections made by the model. The similar statistical bias in this prediction demonstrates that the model can provide insights at least consistent with prior domain expertise. In contrast, when the fingerprint baseline approach is used, the baseline model matches this substructure less frequently and with no discriminating power between positive and negative examples (19.1 % vs 23.9 %).

**Evaluation of Rationales using Synthetic Experiments** Here, we evaluate how often our model would capture target substructures if those substructures were a sufficient indicator of our target task, toxicity. Table 3 summarizes the results, and shows that our model can reliably identify them. Examples of the generated rationales can be seen in Figure 3.

The baseline approach matches fewer instances of the aromatic diazo and polyhalogenation motifs, but does identify more of the aromic nitro groups. Substructures symmetrically centered around a single atom – as in the nitro group – directly correspond to a fingerprint index and are well-described by the baseline model. Correlation between adjacent atoms can cause the RL-CNN model to make an incomplete selection; that is, some of the initial atom features implicitly contain information about its neighbors, which leads to these neighbors appearing less important as rationales. Simplifying the initial atom featurization to decrease this correlation causes the model to successfully select the aromatic nitro group in 17/17 cases.

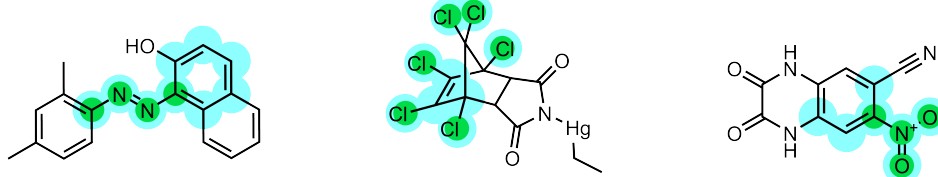

Figure 3: From left to right, example rationales generated for the dataset altered based on the presence of aromatic diazo group, polyhalogenation, and aromatic nitro group. The selected atoms are highlighted in large light blue circles; the predefined toxicophores are highlighted in small green circles.

This confirms that while fingerprint models can do well when the relevant features happen to coincide with a fingerprint index, our rationale model is superior when the relevant features are less well-captured by the exact features of the fingerprint.

## 6 Conclusion

We present a model that treats the problem of selecting rationales from molecules as a reinforcement learning problem. By creating an auxiliary prediction network, we use a learned reward structure to facilitate the selection of atoms in the molecule that are relevant to the prediction task, without significant loss in predictive performance. In this work, we explore the applicability of rationales in the chemistry domain. Through various experiments on the Tox21 and hERG datasets, we demonstrate that our model successfully learns to select important substructures in an unsupervised manner, requiring the same data as an end-to-end prediction task, which is relevant to many applications including drug design and discovery. Molecules are far more complicated to reason about as compared to images or text due to complex chemical theories and a lack of definitive ground truth rationale labels. As deep learning algorithms continue to permeate the chemistry domain, it will be ever more important to consider the interpretability of such models.

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
