# OpenReview forum: "Using Deep Reinforcement Learning to Generate Rationales for Molecules"
_ICLR.cc/2018/Conference — Reject_

### Official Review · AnonReviewer2 · 2017-11-09

**Rating:** 5
**Confidence:** 4

**Review:**


The paper proposes a feature learning technique for molecular prediction using reinforcement learning. The predictive model is an interesting two-step approach where important atoms of the molecule are added one-by-one with a reward given by a second Q-network that learns how well we can solve the prediction problem with the given set of atoms. The overall scheme is intuitive, but

The model is experimented on two small datasets of few thousand of molecules, and compared to a state-of-the-art DeepTox, and also to some basic baselines (RF/SVM/logreg). In the Tox21 dataset the proposed sparse RL-CNN method is less accurate than DeepTox or full CNN. In the hERG dataset RL-CNN is again weaker than the full CNN, but also seems to be beaten by several baseline methods. Overall the results are surprisingly weak, since e.g. with LASSO one often improves by using less features in complex problems. Both datasets should be compared to LASSO as well.

It's somewhat odd that the test performance in table 2 is often better than CV performance. This feels suspicious, especially with 79.0 vs 84.3. The table 2 does not seem reliable result, and should use more folds and more randomizations, etc.

The key problem of the method is its seeming inabability to find the correct number of atoms to use. In both datasets the number of atoms were globally fixed, which is counter-intuitive. The authors should at least provide learning curves where different number of atoms are used; but ideally the method should learn the number of atoms to use for each molecule.

The proposed Q+P network is interesting, but its unclear how well it works in general. There should be experiments that compare the the Q+P model with incresing number of atoms against a full CNN, to see whether the Q+P can converge to maximal performance.

Overall the method is interesting and has a clear impact for molecular prediction, however the paper has limited appeal to the broader audience. Its difficult to assess how useful the Q/P-network is in general. The inability to choose the optimal number of atoms is a major drawback of the method, and the experimental section could be improved. This paper also would probably be more suitable for a chemoinformatics journal, where the rationale learning would be highly appreciated.

---

> ### Author Response · Authors · 2017-12-20
> **Comment**
>
> Thank you for your review of our work.
>
> To address your points:
>
> 1. In molecular problems, often the result is impacted, in part, by properties of the whole molecule, so it is not surprising that using only a subset of the atoms in the molecule will see a decrease in performance. Even in the case of text, we see that using only a partial selection of text reduces the accuracy of prediction (Lei et al. 2016). The main focus of the work was to illustrate that it is possible to extract meaningful rationales through this method.
>
> 2. The reason why the test performance is strictly better than the CV performance is because, as stated in the paper, the two sets of data came from different sources. This is how the original paper used this dataset, so to make a fair comparison, we did the same.
>
> 3. We do see that increasing the number of atoms selected does allow the model to converge to the performance of the full model (as the two models essentially collapse into the same one), but it is a limitation of the model that we have to select, as a hyper parameter, the number of atoms to choose for molecules

---

### Official Review · AnonReviewer3 · 2017-11-27
**Careful discussions would be needed to show that neural nets are good for 'hard' combinatorial problems**

**Rating:** 5
**Confidence:** 4

**Review:**

This paper presents an interesting approach to identify substructural features of molecular graphs contributing to the target task (e.g. predicting toxicity). The algorithm first builds two conv nets for molecular graphs, one is for searching relevant substructures (policy improvement), and another for evaluating the contribution of selected substructures to the target task (policy evaluation). These two phases are iterated in a reinforcement learning manner as policy iterations. Both parts are based on conv nets for molecular graphs, and this framework is a kind of 'self-supervised' scheme compared to the standard situations that the environment provides rewards. The experimental validations demonstrate that this model can learn a competitive-performed conv nets only dependent on the highlighted substructures, as well as reporting some case study on the inhibition assay for hERG proteins.

Technically speaking, the proposed self-supervised scheme with two conv nets is very interesting. This demonstrates how we can perform progressive substructure selections over molecular graphs to highlight relevant substructures as well as maximizing the prediction performance. Given that conv nets for molecular graphs are not trivially interpretable, this would provides a useful approach to use conv nets for more explicit interpretations of how the task can be performed by neural nets.

However, at the same time, I had one big question about the purpose and usage of this approach. As the paper states in Introduction, the target problem is 'hard selection' of substructures, rather than 'soft selection' that neural nets (with attention, for example) or neural-net fingerprints usually provide. Then, the problem would become a combinatorial search problem, which has been long studied in the data mining and machine learning community. There would exist many exact methods such as LEAP, CORK, and graphSig under the name of 'contrast/emerging/discriminative' pattern mining exactly developed for this task. Also, it is widely known that we can even perform a wrapper approach for supervised learning from graphs simultaneously with searching all relevant subgraphs as seen in Kudo+ NIPS 2004, Tsuda ICML 2007, Saigo+ Machine Learning 2009, etc. It would be unconvincing that the proposed neural nets approach fits to this hard combinatorial task rather than these existing (mostly exact) methods.

In addition to the above point, several technical points below would also be unclear.

- A simple heuristic by adding 'selected or not' variables to the atom features works as intended? Because this is fed to the conv net, it seems we can ignore this elements of features by tweaking the weight parameters accordingly. If the conv net performs the best when we use the entire structure, then learning might be forced to ignore the selection. Can we guarantee in some sense this would not happen?

- Zeroing out the atom features also sounds quite simple and a bit groundless. Confusingly, the P network also has an attention mechanism, and it is a bit unclear to me what was actually worked.

- In the experiments, the baseline is based on LR, but this would not be fair because usually we cannot expect any linear relationship for molecular fingerprints. It's highly correlated due to the inclusion relationships between subgraphs. At least, any nonlinear baseline (e.g. Random forest or something?) should be presented for discussing the results.

Pros:
- interesting self-supervised framework provided for highlighting relevant substructures for a given prediction task
- the hard selection setting is encoded in input graph featurization

Cons:
- it would be a bit unconvincing that identifying 'hard selection' is better suited for neural nets, rather than many existing exact methods (without using neural networks). At least one of the typical ones should be compared or discussed.
- I'm still not quite sure whether or not some heuristic parts work as intended.

---

> ### Author Response · Authors · 2017-12-20
> **Comment**
>
> Thank you for your review of our work.
>
> To address your points:
>
> 1. One of the advantages of our model, which is relevant to the chemistry problem, is that we directly account for interactions between different groups of atoms. The exact methods you bring up all try to do some search over the space of the subgraphs in the dataset, picking out the most important ones. However, these selections do not seem to directly incorporate the competing/augmenting effects of having different subgraphs within a molecule. Some of the earlier papers such as (Kudo et al, 2014) use very small datasets, and a CNN can already improve on the performance of the prediction task. None of these methods seem to have any quantitative evaluation of the subgraph features selected, which was what we tried to focus on in our work.
>
> 2. The atom features are zero'd out for the P network only, which is akin to other hard selection problems, in which words that are not selected as rationale are not considered for the final prediction problem. The simple heuristic by adding "selected or not" feature is only for the Q network, which assigns the Q-values for individual atoms, and not part of the prediction networks, which takes the molecule as given, zeros out atoms not selected, and predicts based on that.

---

### Official Review · AnonReviewer1 · 2017-12-12
**Interesting but still immature approach**

**Rating:** 5
**Confidence:** 4

**Review:**

In this manuscript, the authors propose an interesting deep reinforcement learning approach via CNNs to learn the rationales associated to target chemical properties. The paper has merit, but in its current form does not match the acceptance criteria for ICLR.

In particular, the main issue lies in the poor performance reached by the systems, both overall and in comparison with baseline methods, which at the moment hardly justifies the effort required in setting up the DL framework. Moreover, the fact that test performances are sometimes (much) better than training results are quite suspicious in methodological terms.
Finally, the experimental part is quite limited (two small datasets), making it hard to evaluate the scalability (in all sense) of the proposed solution to much larger data.

---

> ### Author Response · Authors · 2017-12-20
> **Comments**
>
> Thank you for your review of our work.
>
> To address your points:
>
> 1. The focus of the model was on the rationale aspect and less on the actual performance of the model. The reason that the test performance on one of the datasets (hERG) is better than cross validation performance is because, as stated in the paper, the two sets of data came from different papers, which was done by the original paper from which the dataset came from.
>
> 2. Most of the publicly available toxicity datasets are very small, so in the chemical context, this is often the best that can be done.

---

### Decision · Program_Chairs · 2018-01-29
**ICLR 2018 Conference Acceptance Decision**

**Decision:**

Reject

**Comment:**


Pro:
 - Interesting approach to tie together reinforcement Q-learning with CNN for prediction and reward function learning in predicting downstream effects of chemical structures, while providing relevant areas for decision-making.

Con:
- Datasets are small, generalizability not clear.
- Performance is not high (although performance wasn't the goal necessarily)
- Sometimes test performance is higher than training performance, making results questionable.
- Should include comparison to other wrapper-based combinatorial approaches.
- Too targeted an appeal/audience (better for chemical journal)